# Transmission of Porcine Circovirus 3 (PCV3) by Xenotransplantation of Pig Hearts into Baboons

**DOI:** 10.3390/v11070650

**Published:** 2019-07-16

**Authors:** Luise Krüger, Matthias Längin, Bruno Reichart, Uwe Fiebig, Yannick Kristiansen, Carolin Prinz, Barbara Kessler, Stefanie Egerer, Eckhard Wolf, Jan-Michael Abicht, Joachim Denner

**Affiliations:** 1Robert Koch Institute, HIV and other retroviruses, 13353 Berlin, Germany; 2Department of Anaesthesiology, Ludwig-Maximilians-Universität München, 81377 Munich, Germany; 3Walter Brendel Centre of Experimental Medicine, Ludwig-Maximilians-Universität München, 81377 Munich, Germany; 4Molecular Animal Breeding and Biotechnology, Gene Center, Ludwig-Maximilians-Universität München, 85764 Oberschleißheim, Germany; 5Robert Koch Institute, Robert Koch Fellow, 13353 Berlin, Germany

**Keywords:** porcine circovirus 3 (PCV3), pigs, non-human primates, xenotransplantation

## Abstract

Porcine circovirus 3 (PCV3) is a newly described member of the virus family Circoviridae. PCV3 is highly distributed among pigs and wild boars worldwide. A sudden introduction of PCV3 was recently observed in a herd of triple genetically modified pigs generated for xenotransplantation. These animals were used as donor pigs for orthotopic heart transplantation into baboons. In four cases, PCV3-positive hearts were transplanted, and transmission of PCV3 to the recipient was observed. PCV3 was found in all organs of the recipient baboons and a higher virus load was found in animals with a longer survival time of the transplant, indicating replication of the virus. This is the first report showing trans-species transmission of PCV3 to baboons by transplantation of a heart from a PCV3-positive donor pig. Sequence analysis showed that PCV3a and PCV3b were present in the infected pigs and were transmitted. Experiments to infect human 293 cells with PCV3 failed.

## 1. Introduction

Porcine circoviruses (PCV) belong to the genus *Circovirus* of the family Circoviridae [1]. Circoviruses are non-enveloped spherical particles with a single-stranded circular small DNA genome, they are the smallest viruses found to be replicating in mammalian cells and two types of PCV have been well characterized in the past, PCV1 and PCV2 [2]. PCV1 was isolated from a pig kidney cell culture (PK15 cells) and found to be apathogenic in pigs, whereas PCV2 was found to be associated with a complex of diseases called PCV-associated disease (PCVAD) or PCVD. Previously the most severe PCVD was called postweaning multisystemic wasting syndrome (PWMS), now called PCV2-systemic disease (PCV2-SD). In addition, PCV2 is associated with PCV2-subclinical infection (PCV2-SI), PCV2-reproductive disease (PCV2-RD), and porcine dermatitis and nephropathy syndrome (PDNS). The finding of PCV2 in subclinical infections suggests that co-factors such as co-infection with other viruses, for example porcine reproductive and respiratory syndrome virus (PRRSV), are required for the induction of diseases. Since both PCV1 and PCV2 can infect human cells [3,4] they may pose a risk when PCV-positive pig organs are used in xenotransplantation [5]. Organs from PCV2 infected pigs may be not fully functional even if the pig is not diseased. 

Xenotransplantation, e.g., the transplantation of pig cells, tissues, or organs, is a technology which was developed to alleviate the shortage in human transplants; in preclinical trials transplanting pig organs into non-human primates, remarkable survival times of the xenotransplants have been observed [6,7]. When evaluating the potential risk posed by PCV1 and PCV2, it is important to know that vaccines against a rotavirus which were contaminated with PCV1 and PCV2 did not induce an infection in the mostly juvenile vaccinated human subjects (for review see [5]). When Göttingen minipigs were analyzed, PCV2 was found in 14% of the animals [8]. Göttingen minipigs have been used as donor pigs for islet cells in a preclinical trial transplanting these cells into cynomolgus monkeys [9]. However, PCV2 transmission was not studied in this preclinical trial. Genetically modified pigs used for islet cell xenotransplantation into marmosets [10], and pigs generated for pig heart transplantation into baboons [11], had been vaccinated against PCV2 and therefore no transmission to the recipients was observed [12].

PCV3 was detected only recently, but it has been shown that the virus is distributed worldwide in farmed pigs [13], as well as in wild boars [14,15,16]. PCV3 was found in healthy animals as well as in animals suffering from different diseases, suggesting that co-infections with other viruses is necessary for the pathogenic potential of PCV3 [13]. As a matter of fact, co-infections with different viruses such as PCV2, PRRSV, and others have been reported.

Here we report the sudden appearance of PCV3 in a herd of triple genetically modified pigs generated for heart xenotransplantation. Hearts from four of the PCV3-positive animals had been transplanted into baboons. Analyzing the baboons revealed PCV3 in all organs of the animals. The longer the survival time of the recipient baboon the higher the virus load, indicating replication of PCV3 in the transplant and/or the baboon. Attempts to infect human 293 cells, however, failed.

## 2. Materials and Methods 

### 2.1. Orthotopic Pig Heart Transplantation 

Transplantation of pig hearts was previously described in detail [11]. Briefly, hearts from α1,3-galactosyltransferase-knockout (GT-KO) pigs that expressed human membrane cofactor protein (CD46) and human thrombomodulin (hTM) were transplanted. All baboons received maintenance immunosuppression based on mycophenolate mofetil, CD40/CD40L costimulation blockade (monkey-specific anti-CD40 monoclonal antibody or PASylated αCD40L Fab), and corticosteroids in addition to an induction therapy with an anti-CD20 antibody and anti-thymocyte-globulin as described in detail [11]. The recipient baboons were weaned from cortisone at an early stage and received antihypertensive treatment since pigs have a lower systolic blood pressure than baboons. In addition, a temsirolimus medication was used to counteract cardiac overgrowth. Baboons A and F were euthanized in good general conditions at day 90, baboons B and C after 6 months, according to the study protocol [11]. 

Both the generation of transgenic animals, as well as interventions on re-cloned animals, were performed with permission of the local regulatory authority, Regierung von Oberbayern (ROB), Sachgebiet 54, 80534 München (approval numbers, AZ 55.2-1-54-2532-70-12, 20 November 2012 and AZ 55.2-1-54-2532-163-14). Applications were reviewed by the ethics committee according to §15 TSchG German Animal Welfare Act. The xenotransplantation experiment was approved by the Government of Upper Bavaria, Munich, Germany (reference number 55.2-1-54-2532-184-2014, September 2015). Housing, feeding, environmental enrichment, and steps taken to minimize suffering, including the use of anesthesia and method of sacrifice, was in accordance with the recommendations of the Weatherall report “*The use of non-human primates in research*”.

### 2.2. Real-Time PCR 

Quantitative real-time PCR was used to detect PCV3 genomes in the isolated DNA of the donor pigs and recipient baboons [16]. The real-time PCR was performed using specific primers and a probe (Table 1). In parallel porcine or baboon glyceraldehyde 3-phosphate dehydrogenase (GAPDH) was determined using specific porcine primers and probes as well as a suitable human primer pair and probe (Table 1). 

The real-time PCR mixture contained 10 µL of SensiFAST Probe No-Rox Mix (Bioline, Luckenwalde, Germany), 300 nM of each primer, 150 nM of the probe, an adequate volume of template DNA (150–300 ng), and sterile distilled water to bring the final volume to 20 µL per sample. For amplification the Stratagene Mx3000P thermal cycler instrument (Agilent Technologies, Santa Clara, CA, USA) was used with the following conditions: denaturation at 95 °C for 5 min and 45 cycles of amplification with denaturation at 95 °C for 30 s, annealing at 55 °C for 1 min, and extension at 72 °C for 45 s. The copy number was quantified using a PCV3 standard containing serial dilutions of a PCV3 open reading frame 2 (ORF2) plasmid, which was generated by cloning part of the PCV3 ORF2 sequence into a pCR 4 TOPO vector as described [16].

PCV1, PCV2 [8], porcine cytomegalovirus (PCMV) [9,21,22], porcine lymphotropic herpesviruses -1, -2, -3 (PLHV-1, -2 and -3) [23] as well as hepatitis E virus) (HEV [24] were detected by PCR or RT-PCR as described previously.

### 2.3. Sequencing

In preparation of the sequencing, DNA was isolated from serum samples of the PCV3-positive donor pigs using the NucleoSpin Virus Kit (Macherey-Nagel, Berlin, Germany). To maximize the amount of viral DNA, rolling circle amplification (RCA) was performed prior to amplification of the PCR products for sequencing. For RCA the TempliPhi amplification Kit (GE Healthcare, Chalfont St Giles, Buckinghamshire, United Kingdom) was used, according to the manufacturer’s protocol and applying 1 µL of isolated DNA. Afterwards, three partially overlapping amplicons were produced using a high fidelity PfuUltra II Fusion HS DNA polymerase (Agilent Technologies, Santa Clara, California, USA) and three different primer-sets (Sequencing 2, 5 and 3 from Table 1). For amplification, the following thermal profile was used: 95 °C for 2 min and 40 cycles of 95 °C for 20 s, 55 °C for 30 s, and 72 °C for 1 min; and a final extension step of 72 °C for 3 min. PCR products were verified by agarose gel electrophoresis and sequenced using the same sequencing primer sets and the BigDye 3.1 Sequencing Kit (Thermo Fischer Scientific, Waltham, MA, USA). 

### 2.4. Infection Experiments with Human Cells

Peripheral blood mononuclear cells (PBMCs) from PCV3-positive pigs were purified using Ficoll gradient centrifugation. 1 × 10^6^ PBMCs per well of a 12-well plate were co-cultured with 3000 human embryonic kidney 293T cells (American Type Culture Collection, ATCC, Manassas, VA, USA, CRL-1573 ( seeded the day before in Dulbecco’s modified Eagle’s medium (DMEM) containing 10% fetal calf serum (FCS) and stimulated with 80 µg/mL phytohemagglutinin (PHA) (Oxoid, Wesel, Germany) at 37 °C. This protocol is based on previous publications showing activation of PCV2 [4,25,26,27], PCMV [28], and porcine endogenous retroviruses (PERV) [29] in mitogen-stimulated pig PBMCs. The co-cultures were split every three days, 7 times in total. After 35 days the still PCV-3 positive supernatant was added to new 50–60% confluent 293T cells and 8 µg/mL polybrene were added. The 293T cells were split 6 times using a trypsin/EDTA solution (0.05%/0.02%, *w*/*v*, Biochrom GmbH, Berlin) during 22 days and cells were collected each time and tested for PCV3. 

## 3. Results

### 3.1. Detection and Quantification of PCV3 in Pigs Generated for Xenotransplantation

Triple genetically modified pigs, i.e., α1,3-galactosyltransferase-knockout (GT-KO) pigs that express human membrane cofactor protein (CD46) and human thrombomodulin (hTM), generated for xenotransplantation of pig hearts were screened for different viruses such as PCMV, HEV, PLHV-1, -2, -3, PCV1, PCV2, and PCV3. All donor pigs were found to be positive for PLHV-1 and -2 using a PCR method, but negative for HEV using a RT-PCR method, and negative for PLHV-3 using a PCR method. All donor pigs were found to be negative for PCV1 and PCV2 using a PCR specific for PCV1 and PCV2. Using real-time PCR specific for PCV3, all pigs of the herd used as donors for heart transplantation were also found negative for PCV3 until March 2018. Starting with a transplantation in March 2018, the first PCV3-positive donor pig which was used for transplantation, was identified (#5803) (Table 2).

In addition, three other pigs (#5807, #6249 and #6253), used for the three subsequent transplantations were also found to be PCV3-positive (Table 2). After the transplantation in October 2018 all further donor pigs used for transplantation were found to be negative again (Table 2, #6329). Despite this, in March 2019, when seven animals taken from the herd of animals produced for xenotransplantation were analyzed, two animals in the herd (#5870 and #6087) were found to be positive, indicating that the virus was still circulating in the herd (Table 3). 

### 3.2. Detection and Quantification of PCV3 in Recipient Baboons After Transplantation of Pig Hearts

When the four donor pigs and the four recipient baboons of the PCV3-positive pig hearts were analyzed, all were found to be PCV3-positive in different organs (Figure 1).

Whereas the virus load was nearly identical in all four donor pigs, differences in the virus load were found in the recipient baboons. In baboons B and C high copy numbers of PCV3 were found in all tested organs as well as in the removed pig heart at the end of the study. In the organs of baboons D and E, the virus load of PCV3 was negative in two independent measurements (Figure 1). In a third measurement, the results in the organs of animals D and E were slightly above the sensitivity of the detection method (not shown). The short survival times of baboons D and E compared with those of baboons B and C seemed to be due to the short survival time of the transplant. The correlation between the survival time of the animal and the amount of PCV3 in the organs of the transplanted baboon indicates that the virus was replicating (Figure 2). Analyzing organs from baboons B and C, very high virus loads were found in the spleen (baboon C) and liver (baboon B). The virus load in the other organs of baboon B and C was also high (Figure 1). 

### 3.3. Subtype Characterisation of PCV3 in the Pigs and Baboons

Parts of the virus genome were amplified and sequenced. The diagnostic amino acids identified by Fux et al. [16], which allow for discriminating between PCV3a and PCV3b, indicate that both PCV3a and PCV3b were present in the donor pigs (Figure 3). The PCV3 subtype found in the donor pig was also detected in the recipient baboon (Figure 3). Although no PCV3-positive DNA could be isolated from organs of the baboons D and E, these animals were PCV3-positive and the subtype in the donor heart was the same as in the pig, PCV3a1 in the case of baboon E and PCV3b1 in the case of baboon D.

### 3.4. Absence of Infection of Human Cells with PCV3

In order to analyze whether PCV3 is able to infect human cells, PBMCs from PCV3-positive pigs were stimulated with the T-cell mitogen phytohemagglutinin (PHA) and co-cultured with human 293T cells. The cells were screened periodically for a PCV3 infection using a real-time PCR. This procedure was chosen because mitogen-stimulation has been shown to increase significantly the replication of PCV2 [4,25,26,27], PCMV [28], and PERV [29]. After 57 days of culture and 13 times of splitting of the 293T cells, no PCV3 sequences were detected, indicating that 293 cells could not be infected with PCV3.

## 4. Discussion

This is the first report showing trans-species transmission of PCV3 to baboons by the transplantation of a heart from a PCV3-positive donor pig. The correlation between the survival time of the transplant in the recipient and the virus load in the organs of the transplanted baboon suggests that the virus was replicating in the animals. PCV3 was found in the removed pig heart after the end of the study as well as in all organs of the baboons. The highest virus load was found in the liver or spleen of the baboons (Figure 1). It remains still unclear whether PCV3 infects baboon cells or whether the replication is only ongoing in the pig heart and the virus is distributed in the baboon with virus-producing pig cells or as free virus in the blood stream. Incubating mitogen-stimulated pig PBMCs from PCV3-positive pigs with human 293 cells as target cells did not result in infection of the 293 cells. Mitogen stimulation was used, because in several publications it has been shown that the related PCV2 was stimulated in mitogen-treated PBMCs [4,25,26,27]. The T-cell mitogen concanavalin A (ConA) enhanced PCV2 replication not only in vitro, but also in lymphoid tissues in vivo [25]. Treatment with IL-2, ConA, and D-glycosamine increased the PCV2 yield more effectively than other treatments [26]. Furthermore, ConA together with methyl-beta-cyclodextrin (MβCD) and D-glucosamine also increased the PCV2 replication in pig kidney cells (PK15) [26]. Here we used another T-cell mitogen, PHA, which also simulates an immune response of the T lymphocytes, for the stimulation of PBMCs. An increased replication after mitogen-stimulation of pig PBMCs was shown not only for PCV2 [4,25,26,27], but also for PCMV [28] and PERV [29]. The fact that human 293 cells were not infected with PCV3 in this first experiment does not mean that human cell cannot be infected with PCV3. There may be several reasons for the negative result, among them a low amount of virus released from PBMCs. To analyze whether PCV3 can infect human cells, the human kidney 293 cell line was used, which is lacking several restriction factors [30] and which has been shown to be highly susceptible for PERVs [31]. In contrast, PCV2 was able to infect 12 different human cell lines, however, the infection efficiency of PCV2 was lower in human cells than in pig PK-15 cells, suggesting that PCV2 infection was limited in human cells [4].

Both PCV3 subtypes, PCV3a and PCV3b, were found in the xenotransplantation donor pigs and obviously both PCV3a and PCV3b were transmitted to the recipient baboons which were found positive for PCV3 by real-time PCR. In the case of PCV3a this was confirmed by sequencing the virus sequences found in the baboon (Figure 3). This was not confirmed for PCV3b, because the virus load in the isolated DNA from the baboons was too low for amplification and sequencing. PCV3a and PCV3b have been found worldwide in farm pigs as well as in wild boars [13,16,18], and by phylogenetic analysis specific nucleotide and amino acid marker positions were identified, which may serve for easy and fast intraspecies classification and genotyping of PCV3 strains [18]. As reported previously, no correlation between the PCV3 variants with their geographical origin was evident [16,18].

PCV3 is considered a putative cause of reproductive failure, encephalitis and myocarditis in perinatal piglets, porcine dermatitis and nephropathy syndrome, and periarteritis in swine in the United States [32], and porcine dermatitis and nephropathy in China [33]. Furthermore, PCV3 has a potential association with swine respiratory disease and diarrhea [34]. However, since PCV3 was found in also in healthy pigs as well as in pigs with numerous severe diseases [13], it seems likely that co-factors such as a second virus infection, an infection with other microorganisms, or genetic factors are required for a pathogenic effect. In the case of PCV2 additional factors also seem to be required for the induction of the overt clinical symptoms. Under experimental conditions, clinical PCVAD is often difficult to reproduce in pigs infected with PCV2 alone [35,36,37,38]. In addition, only a portion of PCV2-infected pigs actually develop the full spectrum of clinical PCVAD, even in the presence of some known co-factors [38]. PCV2 is an immunosuppressive virus, it preferentially targets the lymphoid tissues, which leads to lymphoid depletion and immunosuppression in pigs. PCV2 significantly alters the cytokine responses in infected animals, with IL-10 upregulated, and IL-2 and IL-4 downregulated [39]. Most likely the immunosuppression induced by PCV2 infection predisposes pigs to co-infecting agents. 

It is difficult to evaluate whether, and how, PCV3 contributed to the reduction of the survival time of the transplant. Since the survival time of the transplant with the higher virus load was longer, the effect seems to be low. On the other hand, animals D and E were also infected with PCMV in addition to PCV3 [40]. It is well known that PCMV significantly reduces the survival time of pig kidney and heart transplants in non-human primates [41]. It remains unclear whether the co-infection with PCV3 may have enhanced the pathogenic effect of PCMV or whether the co-infection with PCMV induced pathogenic properties of PCV3. If this was the case, the survival time of the pig xenotransplants may be much longer when using organs from PCV3-free animals. Although the pathogenesis of PCV3 is not fully understood, the observation of trans-species transmissions to a non-human primate raises a concern about the future xenotransplantation studies.

## Figures and Tables

**Figure 1 viruses-11-00650-f001:**
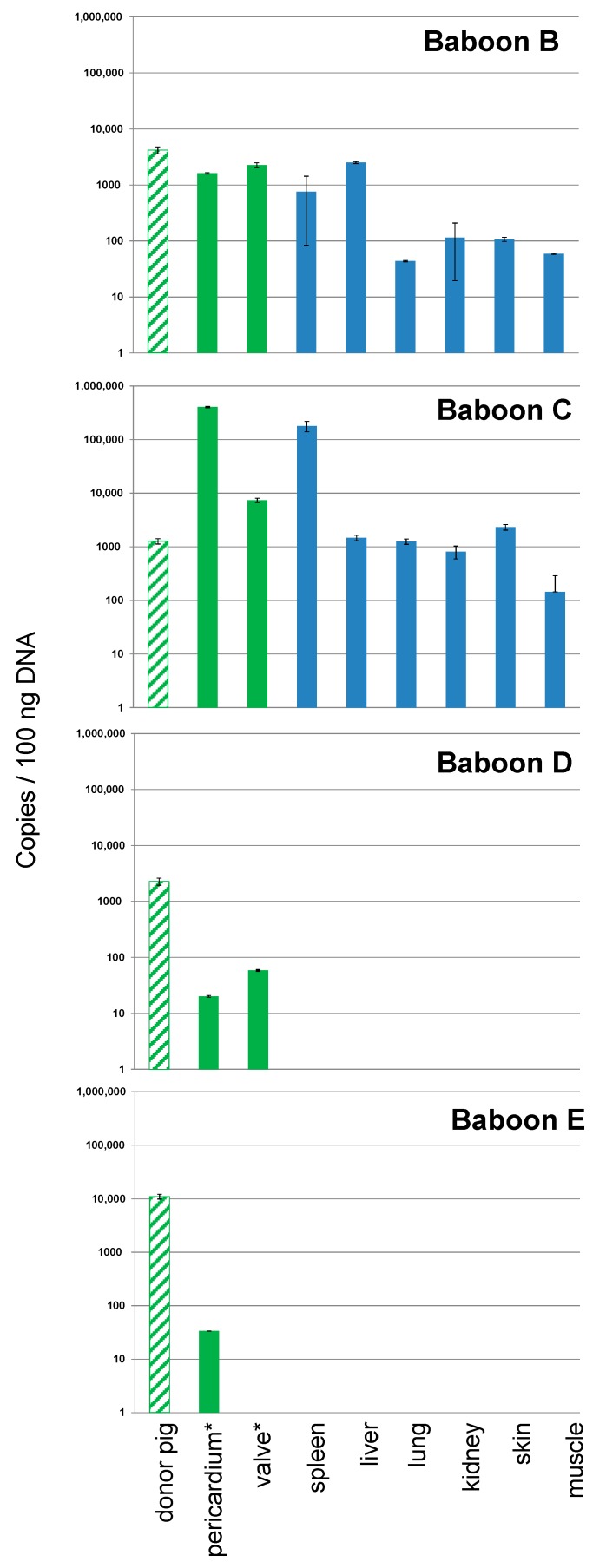
Detection of PCV3 in the organs of four PCV3-positive donor pigs (green hatched), in the transplanted pig heart after its removal at the end of the study (green) and in different organs of the baboon recipient (blue). Column “donor pig”: In the case of baboons B and C the spleens of the donor pigs were analyzed, in the case of baboon D and E the lungs of the donor pigs. Since in the case of baboon D no pericardium and valve material was available, myocardial tissue from the right and left ventricle was tested instead (*). The standard deviation is from two independent measurements.

**Figure 2 viruses-11-00650-f002:**
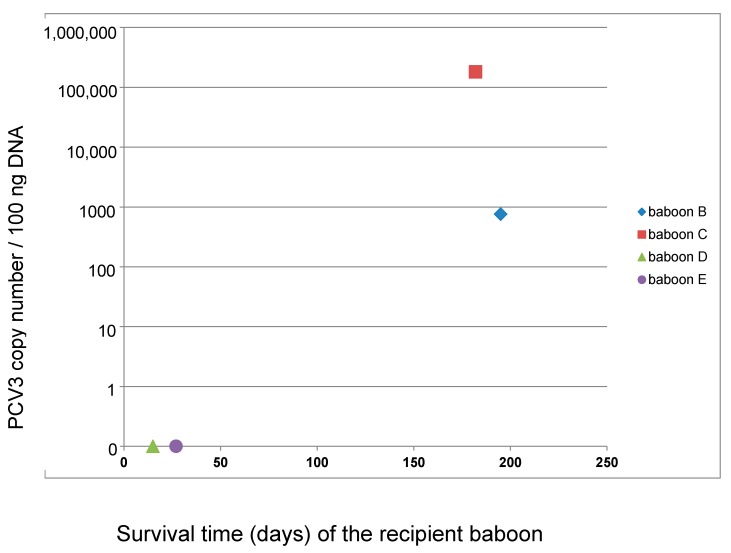
Correlation between the virus load of PCV3 and the survival time of the recipient. The PCV3 copy number is shown for 100 ng DNA in a logarithmic scale which has been modified to show the zero values of baboons D and E.

**Figure 3 viruses-11-00650-f003:**
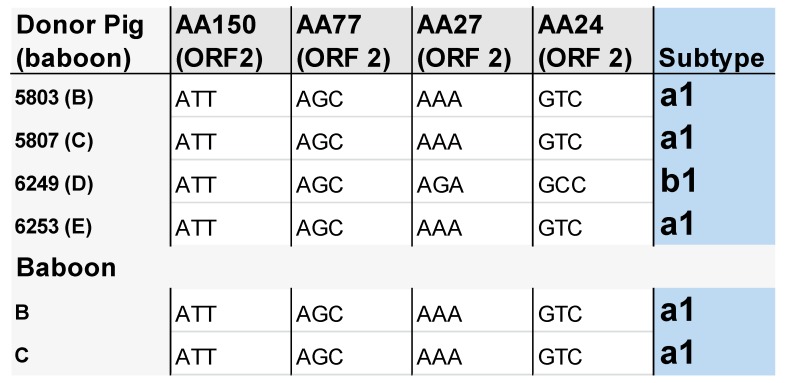
Determination of the subtype of PCV3 in the donor pigs and recipient baboons. No PCV3-positive DNA was available from baboons D and E to perform sequencing. AA, amino acids, ORF 2, open reading frame 2 of PCV3.

**Table 1 viruses-11-00650-t001:** Primers and probes used for PCV screening and sequencing.

Primer Sets	Primer, Probe	Sequences	Accession Number	Position (nt–nt)	Length of the Amplicon (bp)	Reference
PCV3 screening	PCV3 Palinski_For	AGTGCTCCCCATTGAACG	KT869077	1427–1444	135	Palinski et al., 2016 [17]
	PCV3 Palinski_Rev	ACACAGCCGTTACTTCAC		1561–1544		
	PCV3 Palinski_probe	[FAM]-ACCCCATGGCTCAACACATATGACC-[BHQ1]		1473–1449		
Sequencing 2	PCV3 Palinski_For	AGTGCTCCCCATTGAACG		1427–1444	1007	Fux et al., 2018 [18]
	PCV3 Pal Seq2_Rev	CGACCAAATCCGGGTAAGC		433–415		
Sequencing 5	PCV3 Pal Seq1_For	CACCGTGTGAGTGGATATAC		74–93	1072	
	PCV3 Fux 1144_Rev	CACCCCAACGCAATAATTGTA		1144–1124		
Sequencing 3	PCV3 Fux 1137_For	TTGGGGTGGGGGTATTTATT		1137–1156	425	
	PCV3 Palinski_Rev	ACACAGCCGTTACTTCAC		1561–1544		
pGAPDH Set	pGAPDH_For	ACATGGCCTCCAAGGAGTAAGA	n/s	n/s	106	Duvigneau et al., 2005 [19]
	pGAPDH_Rev	GATCGAGTTGGGGCTGTGACT				
	pGAPDH_probe	[HEX]CCACCAACCCCAGCAAGAGCACGC[BHQ1]				
huGAPDH Set	huGAPDH_For	GGCGATGCTGGCGCTGAGTAC	AF261085	365–385	148	Behrendt et al., 2009 [20]
	huGAPDH_Rev	TGGTCCACACCCATGACGA		513–495		
	huGAPDH_probe	[HEX]TTCACCACCATGGAGAAGGCTGGG[BHQ1]		407–430		

PCV, Porcine circoviruses.

**Table 2 viruses-11-00650-t002:** Donor pigs and recipient baboons screened for PCV3.

Date of Transplantation	Nr. Donor Pig	Recipient Baboon	Transplant Survival Time, Days	Presence of PCV3 ^1^
04.10.2017	5528	A	90	no
07.03.2018	5803	B	195	yes
21.03.2018	5807	C	182	yes
10.10.2018	6249	D	15	yes
24.10.2018	6253	E	27	yes
12.12.2018	6329	F	90	no

^1^ PCV3 was tested using a real-time PCR method.

**Table 3 viruses-11-00650-t003:** Detection of PCV3 in animals from the herd in March 2019.

Pig Number	PCV3 ^1^
6493	No
5295	No
4776	No
5806	No
5870	Yes
6086	No
6087	Yes

^1^ PCV3 was tested using a real-time PCR method.

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
