# Peer review of "Transmission of Porcine Circovirus 3 (PCV3) by Xenotransplantation of Pig Hearts into Baboons"

_viruses, 2019, doi:10.3390/v11070650_

Round 1

Reviewer 1 Report

The authors describe the transmission of PCV3 from swine to baboons via transplantation of heart from swine to baboons. This is an extremely important piece of work given that xenotransplation from swine is a valid solution for the shortage of organs. The authors report that out of the six baboon heart recipients, four demonstrated the spread of PCV3 to additional organs. While it may appear that this is a small population size, these are difficult procedures and not necessarily purvey to large numbers. 

Grammatical issues:

Line 163: PCV-3positive should be changed to PCV3 positive

Line 171: The authors report error bars for two independent measurements. Can one truly attain error bars for two measurements? Is this statistically significant?

Author Response

Reviewer 1

Comments and Suggestions for Authors

The authors describe the transmission of PCV3 from swine to baboons via transplantation of heart from swine to baboons. This is an extremely important piece of work given that xenotransplation from swine is a valid solution for the shortage of organs. The authors report that out of the six baboon heart recipients, four demonstrated the spread of PCV3 to additional organs. While it may appear that this is a small population size, these are difficult procedures and not necessarily purvey to large numbers. 

Grammatical issues:

Comment 1

Line 163: PCV-3positive should be changed to PCV3 positive

Answer 1

The typo was corrected.

Comment 2

Line 171: The authors report error bars for two independent measurements. Can one truly attain error bars for two measurements? Is this statistically significant?

Answer 2

The shown error bar demonstrate that there we no large differences between both measurements indicating that the results were correct.

Reviewer 2 Report

The manuscript reported an interesting observation that the newly discovered porcine circovirus 3 (PCV3) was identified in a herd of triple genetically modified pigs for xenotransplantation. For the first time, the authors demonstrated that baboons that had received hearts from the PCV3 positive pigs were infected with PCV3. It is the first time that trans-species transmission of PCV3 via heart xenotransplantation was reported. Although overall the article is very interesting and significant to scientists in this area, there are some concerns that need to be addressed. Appropriately addressing these questions could enhance the significance of this manuscript.

In consideration of that humans and baboons are closely related (~91% DNA similarities), the findings suggested a potential risk of PCV3 transmission in the context of pig-to-human xenotransplantation. In contradictory to the xenotransplantation transmission of PCV3 to baboons, the manuscript reported PCV3 failed to infect human embryonic kidney 293T cells. Does PCV3 infect human cells? It is hard to draw the definite conclusion based on a single experiment using a single human cell line studied in the manuscript. Additional evidence is therefore needed. PCV2, closely related to PCV3, has been reported to be able to infect a board-spectrum of human cells. Although the authors had used an adapted protocol based on previously reported methods for PCV2 infection, it is not clear if the protocol can recapitulate the previous result that PCV2 can infect 293T cell line as reported by Liu et al. (ref # 25; Sci Rep. 2019, 9, 5638). PCV2 infection experiment should be performed in parallel so that a valid positive control for PCV3 infection in human cells is provided. In addition, Liu et al. (ref # 25; Sci Rep. 2019, 9, 5638) had reported PCV2 can infect a variety of human cell lines such as cancerous cell lines A549 and HeLa, and normal cell lines such as HUVEC. More human cell lines, and at least a porcine cell line (e.g. PK15), are recommended to be included in infection experiment of PCV3.

Other concerns are:

i) Line 254 "Nothing is known about the pathogenicity of PCV3."

Accumulating direct and indirect evidence show that PCV3 is associated with porcine diseases. Discussion should be expanded and relevant literatures (some are listed below) should be included.

1.PCV3-associated disease in the United States swine herd, Arruda B et al., Emerging Microbes & Infections, Volume 8, 2019 - Issue 1

"PCV3 is considered a putative cause of reproductive failure, encephalitis and myocarditis in perinatal piglets, porcine dermatitis and nephropathy syndrome, and periarteritis in swine in the United States".

2.Comparative epidemiology of porcine circovirus type 3 in pigs with different clinical presentations. Zhai SL et al., Virol J. 2017 Nov 13;14(1):222

"PCV3 has a potential association with swine respiratory disease and diarrhea."

3.Induction of porcine dermatitis and nephropathy syndrome in piglets by infection with porcine circovirus type 3. Jiang H et al., J Virol. 2019 Feb 5;93(4)

ii) Line 254-256: "Since PCV3 was found in healthy pigs as well as in pigs with numerous severe diseases [11], it seems likely that - similarly to the situation with PCV2 - a second infection is required for a pathogenic effect."

The statement that PCV2 requires a second infection for a pathogenic effect is not accurate. Note that PCV2 it self can lead to lymphoid depletion and immunosuppression in pigs. A second infection can certainly enhance the disease symptom, but is not required for a pathogenic effect. Please see the article below, and other published articles showing PCV2's pathogenesis. 

Porcine circovirus type 2 (PCV2): pathogenesis and interaction with the immune system.

Meng XJ, Annu Rev Anim Biosci. 2013 Jan;1:43-64.

"Porcine circovirus type 2 (PCV2) is the primary causative agent of porcine circovirus-associated disease (PCVAD). The virus preferentially targets the lymphoid tissues, which leads to lymphoid depletion and immunosuppression in pigs. The disease is exacerbated by immunostimulation or concurrent infections with other pathogens."

iii) In the last paragraph, the authors discussed the relation between PCV3 and transplantation survival. The survival time of the animal with the higher virus load was longer, which indicated PCV3 has little effect on survival. However, the increasing virus load, which indicated an established infection in the recipient, is a significant concern by itself. The mechanism that PCV3 establish infection in baboons is likely to be associated to the immunosuppression treatment that the animals received, which provide an environment and enough time for the virus adapting to new host species. Once the adaptation occurred, there is a risk that the virus can use other species as new host species. Although the pathogenesis of PCV3 is not fully understood, this observation should raise a concern for future xenotransplantation studies, and therefore should be highlighted in the discussion. 

Author Response

Reviewer 2

Comments and Suggestions for Authors

The manuscript reported an interesting observation that the newly discovered porcine circovirus 3 (PCV3) was identified in a herd of triple genetically modified pigs for xenotransplantation. For the first time, the authors demonstrated that baboons that had received hearts from the PCV3 positive pigs were infected with PCV3. It is the first time that trans-species transmission of PCV3 via heart xenotransplantation was reported. Although overall the article is very interesting and significant to scientists in this area, there are some concerns that need to be addressed. Appropriately addressing these questions could enhance the significance of this manuscript.

Comment 1

In consideration of that humans and baboons are closely related (~91% DNA similarities), the findings suggested a potential risk of PCV3 transmission in the context of pig-to-human xenotransplantation. In contradictory to the xenotransplantation transmission of PCV3 to baboons, the manuscript reported PCV3 failed to infect human embryonic kidney 293T cells. Does PCV3 infect human cells? It is hard to draw the definite conclusion based on a single experiment using a single human cell line studied in the manuscript. Additional evidence is therefore needed. PCV2, closely related to PCV3, has been reported to be able to infect a board-spectrum of human cells. Although the authors had used an adapted protocol based on previously reported methods for PCV2 infection, it is not clear if the protocol can recapitulate the previous result that PCV2 can infect 293T cell line as reported by Liu et al. (ref # 25; Sci Rep. 2019, 9, 5638). PCV2 infection experiment should be performed in parallel so that a valid positive control for PCV3 infection in human cells is provided. In addition, Liu et al. (ref # 25; Sci Rep. 2019, 9, 5638) had reported PCV2 can infect a variety of human cell lines such as cancerous cell lines A549 and HeLa, and normal cell lines such as HUVEC. More human cell lines, and at least a porcine cell line (e.g. PK15), are recommended to be included in infection experiment of PCV3.

Answer 1

We fully agree with the reviewer that it would be interesting to know whether PCV3 can infect human cells (at least in vitro) and this would enhance the significance of the manuscript. Our attempt to use PBMCs from PCV3-positive animals and human 293 cells failed certainly for many reasons, possibly including the low amount of virus released from the pig PBMCs. We will explain the possible reasons in more detail in the Discussion section.

Since we do not have titrated PCV2 and PCV3 batches, a requested comparative infection assays would be difficult to perform, it would be very time consuming and the result may be unclear. Furthermore, our funding allows us only to focus on detection and elimination of potentially zoonotic viruses in the context of xenotransplantation. Therefore we think that it is important to publish the manuscript in the given form, especially based on what the reviewer said in his last paragraph. We will underline that this infection experiment does not mean that PCV3 does not infect human cells. From the observed transmission of PCV3 to baboons only one conclusion can be drawn: this virus has to be eliminated from the donor pigs.

Comment 2

Other concerns are:

i) Line 254 "Nothing is known about the pathogenicity of PCV3."

Accumulating direct and indirect evidence show that PCV3 is associated with porcine diseases. Discussion should be expanded and relevant literatures (some are listed below) should be included.

1.PCV3-associated disease in the United States swine herd, Arruda B et al., Emerging Microbes & Infections, Volume 8, 2019 - Issue 1

"PCV3 is considered a putative cause of reproductive failure, encephalitis and myocarditis in perinatal piglets, porcine dermatitis and nephropathy syndrome, and periarteritis in swine in the United States".

2.Comparative epidemiology of porcine circovirus type 3 in pigs with different clinical presentations. Zhai SL et al., Virol J. 2017 Nov 13;14(1):222

"PCV3 has a potential association with swine respiratory disease and diarrhea."

3.Induction of porcine dermatitis and nephropathy syndrome in piglets by infection with porcine circovirus type 3. Jiang H et al., J Virol. 2019 Feb 5;93(4)

Answer 2

We changed this sentence, extended the discussion and added the proposed publications (thank you very much):

PCV3 is considered a putative cause of reproductive failure, encephalitis and myocarditis in perinatal piglets, porcine dermatitis and nephropathy syndrome, and periarteritis in swine in the United States [31] and PCV3 has a potential association with swine respiratory disease and diarrhea [32]. However, since PCV3 was found in also in healthy pigs as well as in pigs with numerous severe diseases [12], it seems likely that - similarly to the situation with PCV2 - co-factors such as a second virus infection, an infection with other microorganisms or genetic factors  is are required for a pathogenic effect. In the case of PCV2 also additional factors seem to be required for the induction of the overt clinical symptoms. Under experimental conditions, clinical PCVAD is often difficult to reproduce in pigs infected with PCV2 alone [33-36]. In addition, only a portion of PCV2-infected pigs actually develop the full spectrum of clinical PCVAD, even in the presence of some known co-factors [36]. PCV2 is an immunosuppressive virus, it preferentially targets the lymphoid tissues, which leads to lymphoid depletion and immunosuppression in pigs. PCV2 significantly alters the cytokine responses in infected animals, IL-10 is upregulated, IL-2 and IL-4 downregulated [37]. Most likely the immunosuppression induced by PCV2 infection predisposes pigs to co-infecting agents. 

Comment 3

ii) Line 254-256: "Since PCV3 was found in healthy pigs as well as in pigs with numerous severe diseases [11], it seems likely that - similarly to the situation with PCV2 - a second infection is required for a pathogenic effect."

The statement that PCV2 requires a second infection for a pathogenic effect is not accurate. Note that PCV2 it self can lead to lymphoid depletion and immunosuppression in pigs. A second infection can certainly enhance the disease symptom, but is not required for a pathogenic effect. Please see the article below, and other published articles showing PCV2's pathogenesis. 

Porcine circovirus type 2 (PCV2): pathogenesis and interaction with the immune system.

Meng XJ, Annu Rev Anim Biosci. 2013 Jan;1:43-64.

"Porcine circovirus type 2 (PCV2) is the primary causative agent of porcine circovirus-associated disease (PCVAD). The virus preferentially targets the lymphoid tissues, which leads to lymphoid depletion and immunosuppression in pigs. The disease is exacerbated by immunostimulation or concurrent infections with other pathogens."

Answer 3

Thank you very much for this remark. The situation is indeed very difficult. Many facts indicate that other factors such as other viruses, other microorganisms or genetic factors are required for the induction of the overt clinical symptoms. Under experimental conditions, clinical PCVAD is often difficult to reproduce in pigs infected with PCV2 alone [1-4]. In addition, only a portion of PCV2-infected pigs actually develop the full spectrum of clinical PCVAD, even in the presence of some known co-factors [4]. We changed the sentence accordingly and added additional references.

1.      Opriessnig T, Meng XJ, Halbur PG. Porcine circovirus type 2 associated disease: update on current terminology, clinical manifestations, pathogenesis, diagnosis, and intervention strategies. J Vet Diagn Invest. 2007 Nov;19(6):591-615.

2.      Opriessnig T, Halbur PG. Concurrent infections are important for expression of porcine circovirus associated disease. Virus Res. 2012 Mar;164(1-2):20-32.

3.      Fenaux M, Halbur PG, Haqshenas G, Royer R, Thomas P, Nawagitgul P, Gill M, Toth TE, Meng XJ. Cloned genomic DNA of type 2 porcine circovirus is infectious when injected directly into the liver and lymph nodes of pigs: characterization of clinical disease, virus distribution, and pathologic lesions. J Virol. 2002 Jan;76(2):541-51.

4.      Meng XJ.Porcine circovirus type 2 (PCV2): pathogenesis and interaction with the immune system. Annu Rev Anim Biosci. 2013 Jan;1:43-64.

Comment 4 

iii) In the last paragraph, the authors discussed the relation between PCV3 and transplantation survival. The survival time of the animal with the higher virus load was longer, which indicated PCV3 has little effect on survival. However, the increasing virus load, which indicated an established infection in the recipient, is a significant concern by itself. The mechanism that PCV3 establish infection in baboons is likely to be associated to the immunosuppression treatment that the animals received, which provide an environment and enough time for the virus adapting to new host species. Once the adaptation occurred, there is a risk that the virus can use other species as new host species. Although the pathogenesis of PCV3 is not fully understood, this observation should raise a concern for future xenotransplantation studies, and therefore should be highlighted in the discussion. 

Answer 4

We fully agree with the reviewer that transmission of PCV3 to a non-human primate in a preclinical xenotransplantation trial raise concern for future xenotransplantation studies and we highlighted this in the discussion.

Round 2

Reviewer 2 Report

The authors had addressed most of concerns I had. 

1. The concern was: "Does PCV3 infect human cells? It is hard to draw the definite conclusion based on a single experiment using a single human cell line studied in the manuscript."

In the revised article, the limitations of single experiment in a single human cell line have been clearly discussed.

2. The concern was: Accumulating direct and indirect evidence show that PCV3 is associated with porcine diseases. Discussion should be expanded and relevant literatures (some are listed below) should be included.

The discussion of PCV3's pathogenesis was much improved. I noticed a minor problem with an added reference. Line # 264 [32], #390-2: a wrong reference (Jiang et al.) was referenced for associated diarrhea and respiratory diseases. It should be "Comparative epidemiology of porcine circovirus type 3 in pigs with different clinical presentations. Zhai SL et al., Virol J. 2017 Nov 13;14(1):222". Jiang et al. reported PCV3 and dermatitis and nephropathy. Please correct this citation.

3. The pathogenesis of PCV2.

The authors had added references and improved the discussion. 

4. The significance of the trans-species transmission.

Clearly addressed at the end of the article. 

Author Response

Reviewer 2

The authors had addressed most of concerns I had. 

1. The concern was: "Does PCV3 infect human cells? It is hard to draw the definite conclusion based on a single experiment using a single human cell line studied in the manuscript."

In the revised article, the limitations of single experiment in a single human cell line have been clearly discussed.

2. The concern was: Accumulating direct and indirect evidence show that PCV3 is associated with porcine diseases. Discussion should be expanded and relevant literatures (some are listed below) should be included.

Comment 1

The discussion of PCV3's pathogenesis was much improved. I noticed a minor problem with an added reference. Line # 264 [32], #390-2: a wrong reference (Jiang et al.) was referenced for associated diarrhea and respiratory diseases. It should be "Comparative epidemiology of porcine circovirus type 3 in pigs with different clinical presentations. Zhai SL et al., Virol J. 2017 Nov 13;14(1):222". Jiang et al. reported PCV3 and dermatitis and nephropathy. Please correct this citation.

Answer 1

We corrected the text and included the correct reference,

3. The pathogenesis of PCV2.

The authors had added references and improved the discussion. 

4. The significance of the trans-species transmission.

Clearly addressed at the end of the article.